# 3DGS-DET: Empower 3D Gaussian Splatting with Boundary Guidance and Box-Focused Sampling for 3D Object Detection

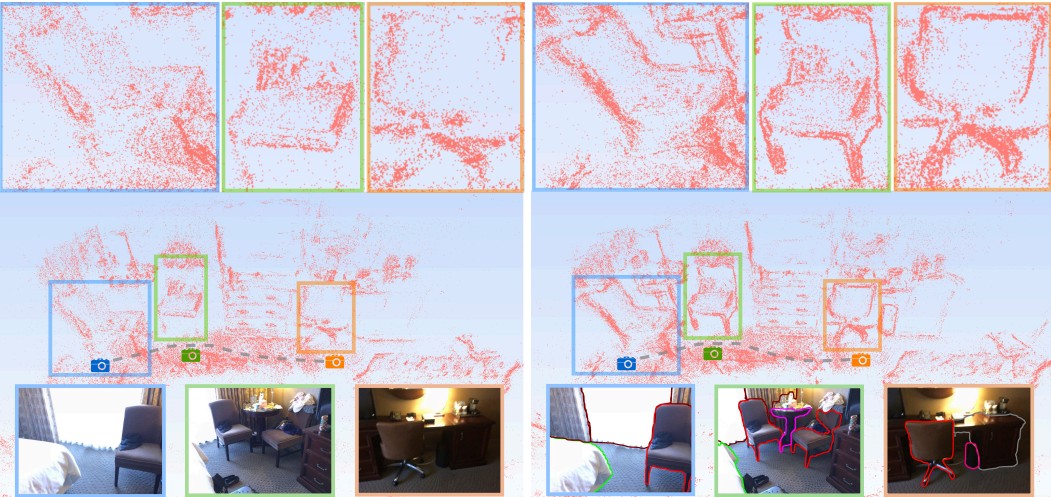

Figure 1: Illustration of the proposed Boundary Guidance. By incorporating Boundary Guidance in the training of 3D Gaussian Splatting (3DGS), we significantly improve the spatial distribution of Gaussian blobs relating objects and the background. To better show this improved spatial distribution, we visualize only the positions of the Gaussian blobs, omitting other attributes for clarity.

## ABSTRACT

Neural Radiance Fields (NeRF) are widely used for novel-view synthesis and have been adapted for 3D Object Detection (3DOD), offering a promising approach to 3D object detection through view-synthesis representation. However, NeRF faces inherent limitations: (i) It has limited representational capacity for 3DOD due to its implicit nature, and (ii) it suffers from slow rendering speeds. Recently, 3D Gaussian Splatting (3DGS) has emerged as an explicit 3D representation that addresses these limitations with faster rendering capabilities. Inspired by these advantages, this paper introduces 3DGS into 3DOD for the first time, identifying two main challenges: (i) Ambiguous spatial distribution of Gaussian blobs – 3DGS primarily relies on 2D pixel-level supervision, resulting in unclear 3D spatial distribution of Gaussian blobs and poor differentiation between objects and background, which hinders 3DOD; (ii) Excessive background blobs – 2D images often include numerous background pixels, leading to densely reconstructed 3DGS with many noisy Gaussian blobs representing the background, negatively affecting detection. To tackle the challenge (i), we leverage the fact that 3DGS reconstruction is derived from 2D images, and propose an elegant and efficient solution by incorporating 2D Boundary Guidance to significantly enhance the spatial distribution of Gaussian blobs, resulting in clearer differentiation between objects and their background (see Fig. 1). To address the challenge (ii), we propose a Box-Focused Sampling strategy using 2D boxes to generate object probability distribution in 3D spaces, allowing effective probabilistic sampling in 3D to retain more object blobs and reduce noisy background blobs. Benefiting from the proposed Boundary Guidance and Box-Focused Sampling, our final method, **3DGS-DET**, achieves significant improvements (**+5.6** on mAP@0.25, **+3.7** on mAP@0.5) over our basic pipeline version, without introducing any additional learnable parameters. Fur-

thermore, 3DGS-DET significantly outperforms the state-of-the-art NeRF-based method, NeRF-Det, achieving improvements of **+6.6** on mAP@0.25 and **+8.1** on mAP@0.5 for the ScanNet dataset, and impressive **+31.5** on mAP@0.25 for the ARKITScenes dataset. We are committed to releasing all codes and data within one month following the paper's acceptance.

# 1 INTRODUCTION

3D Object Detection (3DOD) (Qi et al., 2017a; 2019) is a fundamental task in computer vision, providing foundations for wide realistic application scenarios such as autonomous driving, robotics, and industrial production, as accurate localization and classification of objects in 3D space are critical for these applications. Most existing 3DOD methods (Rukhovich et al., 2022b;a) explored using non-view-synthesis representations, including point clouds, RGBD, and multi-view images, to perform 3D object detection. However, these approaches mainly focus on the perception perspective and lack the capability for novel view synthesis.

Neural Radiance Fields (NeRF) (Mildenhall et al., 2021) provide an effective manner for novel view synthesis and have been adapted for 3D Object Detection (3DOD) through view-synthesis representations (Xu et al., 2023; Hu et al., 2023). However, as a view-synthesis representation for 3D object detection, NeRF has inherent limitations: 1) Its implicit nature restricts its representational capacity for 3DOD, and 2) it suffers from slow rendering speeds. Recently, 3D Gaussian Splatting (3DGS) (Kerbl et al., 2023) has emerged as an explicit 3D representation that offers faster rendering, effectively addressing these limitations. Inspired by these strengths, our work is *the first* to introduce 3DGS into 3DOD. In this exploration, we identify two primary challenges: (i) Ambiguous spatial distribution of Gaussian blobs – 3DGS primarily relies on 2D pixel-level supervision, resulting in unclear 3D spatial distribution of Gaussian blobs and insufficient differentiation between objects and background, which hinders effective 3DOD; (ii) Excessive background blobs – 2D images often contain numerous background pixels, leading to densely populated 3DGS with many noisy Gaussian blobs representing the background, negatively impacting the detection of foreground 3D objects.

To address the above-discussed challenges, we further empower 3DGS with two novel strategies for 3D object detection (i) *2D Boundary Guidance Strategy*: Given the fact that 3DGS reconstruction is optimized from 2D images, we introduce a novel strategy by incorporating 2D Boundary Guidance to achieve a more suitable 3D spatial distribution of Gaussian blobs for detection. Specifically, we first perform object boundary detection on posed images, then overlay the boundaries onto the images, and finally train the 3DGS model. This proposed strategy can facilitate the learning of a spatial Gaussian blob distribution that is more differentiable for the foreground objects and the background (see Fig. 1). (ii) *Box-Focused Sampling Strategy*: This strategy further leverages 2D boxes to establish 3D object probability spaces, enabling an object probabilistic sampling of Gaussian blobs to effectively preserve object blobs and prune background blobs. Specifically, we project the 2D boxes that cover objects in images into 3D spaces to form frustums. The 3D Gaussian blobs within the frustum have a higher probability of being object blobs compared to those outside. Based on this strategy, we construct 3D object probability spaces and sample Gaussian blobs accordingly, finally preserving more object blobs and reducing noisy background blobs.

In summary, the contributions of this work are fourfold:

- To the best of our knowledge, we are the first to integrate 3D Gaussian Splatting (3DGS) into 3D Object Detection (3DOD), representing a novel contribution to the field. We propose **3DGS-DET**, which empowers 3DGS with Boundary Guidance and Box-Focused Sampling for 3DOD.

- We design *Boundary Guidance* to optimize 3DGS with the guidance of object boundaries, which achieves a significantly better spatial distribution of Gaussian blobs and clearer differentiation between objects and the background, thereby effectively enhancing 3D object detection.

- We propose *Box-Focused Sampling*, which establishes 3D object probability spaces, enabling a higher sampling probability to be assigned to object-related 3D Gaussian blobs. This probabilistic sampling strategy preserves more object blobs and suppresses noisy background blobs, therefore producing further improved detection performance.

- With zero additional learnable parameters, Boundary Guidance and Box-Focused Sampling improve detection by **5.6 points** on mAP@0.25 and **3.7 points** on mAP@0.5 as demonstrated in our ablation study. Furthermore, our final approach, 3DGS-DET, significantly outperforms the state-of-the-art NeRF-based method, NeRF-Det, on both ScanNet (**+6.6** on mAP@0.25, **+8.1** on mAP@0.5) and ARKITScenes (**+31.5** on mAP@0.25).

## 2 RELATED WORKS

**3D Gaussian Splatting (3DGS)** is an effective explicit representation that models 3D scenes or objects using Gaussian blobs – small, continuous Gaussian functions distributed across 3D space, enabling faster rendering. Recent works (Shen et al., 2024b; Liu et al., 2024b; Lee et al., 2024) have shown that 3DGS is highly suitable for dynamic scene modeling. Additionally, some studies (Lin et al., 2024; Zhang et al., 2024; Xiong et al., 2024; Wang & Xu, 2024; Liu et al., 2024a; Feng et al., 2024) also demonstrate its efficiency in processing large-scale 3D scene data. A key focus of recent 3DGS research is integrating semantic understanding to enhance perception capabilities. Researchers (Zhou et al., 2024; Qin et al., 2024; Shi et al., 2024; Zuo et al., 2024) leverage advanced 2D foundational models, such as SAM (Kirillov et al., 2023) and CLIP (Radford et al., 2021), along with feature extraction methods like DINO (Zhang et al., 2022), to boost perception effectiveness. Unlike previous methods that often overlook specific challenges of 3D Object Detection (3DOD), our approach uniquely introduces Boundary Guidance and Box-Focused Sampling, marking the first exploration of 3DGS as a representation for the 3D object detection task.

**Non-View-Synthesis Representation-Based 3D Object Detection.** Traditional 3D detection tasks primarily utilize the following representations: (i) Point cloud-based methods (Yang et al., 2018; Ali et al., 2018; Shi et al., 2019; Qi et al., 2019; 2021; Wang et al., 2022b; Peng et al., 2022; Wang et al., 2022a; Rukhovich et al., 2022a; Cao et al., 2023; 2024) directly process unstructured 3D points captured by sensors like LiDAR or depth cameras. Techniques such as VoteNet (Qi et al., 2019) and CAGroup3D (Wang et al., 2022a) efficiently handle point clouds, capturing detailed geometries while facing challenges in computational efficiency due to their irregular structure. Some researches (Zhou & Tuzel, 2018; Ye et al., 2020; Deng et al., 2021; Mao et al., 2021; Noh et al., 2021; Chen et al., 2023b; Mahmoud et al., 2023) divide 3D space into uniform volumetric units, enabling 3D convolutional neural networks to process the data, although they encounter trade-offs between resolution and memory usage. (ii) Multi-view image-based methods (Wang et al., 2022c; Xiong et al., 2023; Wang et al., 2023; Chen et al., 2023a; Feng et al., 2023; Tu et al., 2023; Shen et al., 2024a) leverage 2D images from multiple perspectives to reconstruct 3D structures. (iii) RGB-D based methods (Qi et al., 2018; 2020; Luo et al., 2020) enhance 3D object detection by combining 2D images cues, with 3D data to improve accuracy. However, these representations predominantly focus on perception and lack the capability for novel view synthesis.

**View-Synthesis Representation-Based 3D Object Detection.** Neural Radiance Fields (NeRF) (Mildenhall et al., 2021) have become popular for novel-view-synthesis and have been adapted for 3D Object Detection (3DOD) (Hu et al., 2023; Xu et al., 2023). These adaptations present promising solutions for detecting 3D objects using view-synthesis representations. For instance, NeRF-RPN (Hu et al., 2023) employs voxel representations, integrating multi-scale 3D neural volumetric features to perform category-agnostic box localization rather than category-specific object detection. NeRF-Det (Xu et al., 2023) incorporates multi-view geometric constraints from the NeRF component into 3D detection. Notably, NeRF-RPN focuses on class-agnostic box detection, while NeRF-Det targets class-specific object detection. Our work follows the class-specific setting of NeRF-Det. However, NeRF faces significant challenges: its implicit nature limits its representational capacity for 3D object detection, and it suffers from slow rendering speeds. 3D Gaussian Splatting (3DGS) (Kerbl et al., 2023) has emerged as an explicit 3D representation, offering faster rendering and effectively addressing these limitations. Motivated by these advantages, our work introduces 3DGS into 3DOD for the first time, and presents novel designs to adapt 3DGS for detection, making significant differences from NeRF-based methods (Hu et al., 2023; Xu et al., 2023).

## 3 METHODOLOGY

The pipeline of our 3DGS-DET is illustrated in the bottom row of Fig. 2. Initially, we train the 3D Gaussian Splatting (3DGS) on the input scenes using the proposed Boundary Guidance, which significantly enhances the spatial distribution of Gaussian blobs, resulting in clearer differentiation between objects and the background. Subsequently, we apply the proposed Box-Focused Sampling, which effectively preserves object-related blobs while suppressing noisy background blobs. The sampled 3DGS is then fed into the detection framework for training. In this section, we detail our method step by step. First, we introduce the preliminary concept of 3D Gaussian Splatting (3DGS) in Sec. 3.1. As the first to introduce 3DGS in 3D object detection, we establish the basic pipeline in Sec. 3.2, utilizing 3DGS for input and output detection predictions. We then present Boundary Guidance in Sec. 3.3. Finally, we describe the Box-Focused Sampling Strategy in Sec. 3.4.

Figure 2: Pipeline overview (zooming in for a clearer view). The top row illustrates our basic pipeline detailed in Sec. 3.2. The bottom row shows our 3DGS-DET pipeline with both Boundary Guidance (Sec. 3.3) and Box-Focused Sampling (Sec. 3.4) embedded. The Boundary Guidance can significantly improve the 3D spatial distribution of Gaussian blobs, and thus produce clearer differentiation between objects and the background. The Box-Focused Sampling effectively preserves more object-related blobs while suppressing noisy background blobs, compared to random sampling. These two proposed strategies together largely advance the 3D detection performance.

## 3.1 PRELIMINARY: 3D GAUSSIAN SPLATTING

In our proposed method, 3DGS-DET, the input scene is represented using 3D Gaussian Splatting (3DGS) (Kerbl et al., 2023), formulated as follows:

$$G = \{(\boldsymbol{\mu}_i, \boldsymbol{S}_i, \boldsymbol{R}_i, \boldsymbol{c}_i, \boldsymbol{\alpha}_i)\}_{i=1}^{N}, \tag{1}$$

where $N$ denotes the number of Gaussian blobs. Each blob is characterized by its 3D coordinate $\boldsymbol{\mu}_i$, scaling matrix $\boldsymbol{S}_i$, rotation matrix $\boldsymbol{R}_i$, color features $\boldsymbol{c}_i$, and opacity $\boldsymbol{\alpha}_i$. These attributes define the Gaussian through a covariance matrix $\Sigma = \boldsymbol{R}\boldsymbol{S}\boldsymbol{S}^T\boldsymbol{R}^T$, centered at $\boldsymbol{\mu}$:

$$G(\boldsymbol{x}) = \exp^{\left(-\frac{1}{2}(\boldsymbol{x}-\boldsymbol{\mu})^T\Sigma^{-1}(\boldsymbol{x}-\boldsymbol{\mu})\right)}. \tag{2}$$

During rendering, opacity modulates the Gaussian. By projecting the covariance onto a 2D plane (Zwicker et al., 2001), we derive the projected Gaussian, and utilize volume rendering (Max, 1995) to compute the image pixel colors:

$$C = \sum_{k=1}^{K} \alpha_k c_k \prod_{j=1}^{k-1}(1 - \alpha_j), \tag{3}$$

where $K$ is the number of sampling points along the ray. $\alpha_i$ is determined by evaluating a 2D Gaussian with covariance $\Sigma$, multiplied by the learned opacity (Yifan et al., 2019). The initial 3D coordinates of each Gaussian are based on Structure from Motion (SfM) points (Schonberger & Frahm, 2016). Gaussian attributes are refined to minimize the image reconstruction loss:

$$L_{\text{render}} = (1 - \lambda)L_1(I, \hat{I}) + \lambda L_{\text{D-SSIM}}(I, \hat{I}), \tag{4}$$

where $\hat{I}$ represents the ground truth images. Additional details can be found in Kerbl et al. (2023).

## 3.2 PROPOSED BASIC PIPELINE OF 3DGS FOR 3D OBJECT DETECTION

In this section, we build our basic pipeline by directly utilizing the original 3D Gaussian Splatting (3DGS) for 3D Object Detection (3DOD) without any further improvement. As depicted in the top row of Fig. 2, we train the 3DGS representation of the input scene using posed images, denoted as $G = \{(\boldsymbol{\mu}_i, \boldsymbol{S}_i, \boldsymbol{R}_i, \boldsymbol{c}_i, \boldsymbol{\alpha}_i)\}_{i=1}^{N}$. Given that the number of Gaussian blobs $N$ is too large for them to be input into the detector, we perform random sampling to select a subset of Gaussian blobs, denoted as $\hat{G} = \{(\boldsymbol{\mu}_i, \boldsymbol{S}_i, \boldsymbol{R}_i, \boldsymbol{c}_i, \boldsymbol{\alpha}_i)\}_{i=1}^{M}$, where $M < N$. We then concatenate the attributes of the Gaussian blobs along the channel dimension as follows:

$$\hat{G}_{\text{input}} = \text{Concat}(\boldsymbol{\mu}_i, \boldsymbol{S}_i, \boldsymbol{R}_i, \boldsymbol{c}_i, \boldsymbol{\alpha}_i) \quad \forall i \in \{1, \dots, M\}. \tag{5}$$

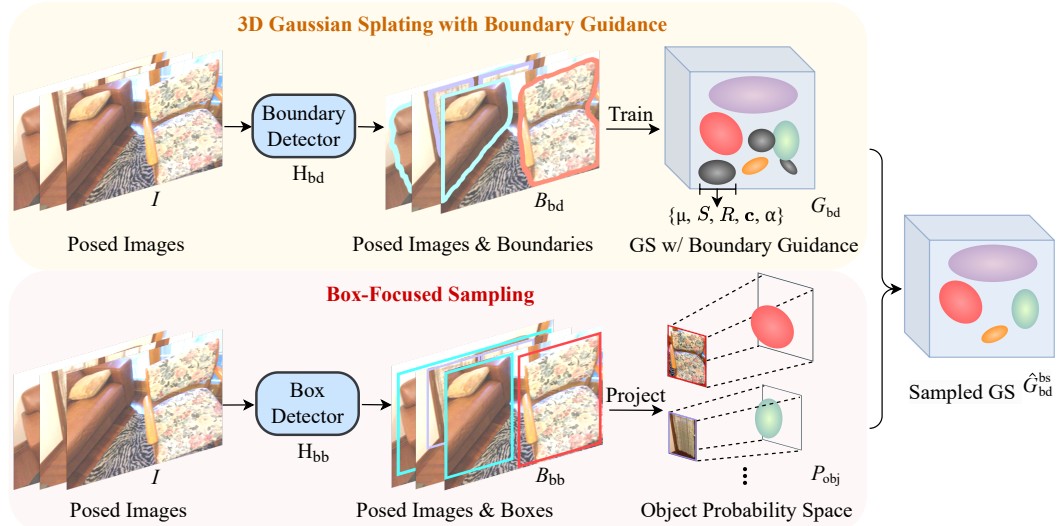

Figure 3: Illustration of the proposed Boundary Guidance and Box-Focused Sampling strategies. In the top row, Boundary Guidance is constructed by three steps, *i.e.,* detecting boundaries on posed images, overlaying them to images, and training a 3DGS model to achieve a more distinct spatial distribution of Gaussian blobs for objects and the background. In the bottom row, Box-Focused Sampling is achieved by conducting object detection on posed images. The predicted 2D boxes are projected into the 3D domain to establish object probability spaces, allowing probabilistic sampling of Gaussians to preserve more object blobs and suppress noisy background blobs.

This concatenated representation $\hat{G}_{\text{input}}$ is then fed into the subsequent detection tool. Note that since 3DGS is an explicit 3D representation, $\hat{G}_{\text{input}}$ can be utilized with any point-cloud-based detector by retraining the detector model on 3DGS representation. In our study, the research focus is on enhancing 3DGS for 3DOD in general, rather than designing a specific detector. Therefore, we utilize the existing work (Rukhovich et al., 2022a) as the detection tool. The final detection predictions are obtained as follows:

$$P = \text{F}(\hat{G}_{\text{input}}) = (\boldsymbol{p}, \boldsymbol{z}, \boldsymbol{b}) , \tag{6}$$

where F denotes the detector tool and $P$ represents the predictions, including classification probabilities $\boldsymbol{p}$, centerness $\boldsymbol{z}$, and bounding box regression parameters $\boldsymbol{b}$.

The training loss (Rukhovich et al., 2022a) is defined as:

$$L_{\text{det}} = \frac{1}{N_{\text{pos}}} \sum_{\hat{x},\hat{y},\hat{z}} \left( \mathbb{1}_{\{p(\hat{x},\hat{y},\hat{z}) \neq 0\}} L_{\text{reg}}(\hat{\boldsymbol{b}}, \boldsymbol{b}) + \mathbb{1}_{\{p(\hat{x},\hat{y},\hat{z}) \neq 0\}} L_{\text{cntr}}(\hat{\boldsymbol{z}}, \boldsymbol{z}) + L_{\text{cls}}(\hat{\boldsymbol{p}}, \boldsymbol{p}) \right), \tag{7}$$

where the number of matched positions $N_{\text{pos}}$ is given by $\sum_{\hat{x},\hat{y},\hat{z}} \mathbb{1}_{\{p(\hat{x},\hat{y},\hat{z}) \neq 0\}}$. Ground truth labels are indicated with a hat symbol. The regression loss $L_{\text{reg}}$ is based on Intersection over Union (IoU), the centerness loss $L_{\text{cntr}}$ uses binary cross-entropy, and the classification loss $L_{\text{cls}}$ employs focal loss. Further details on the detection tool can be found in Rukhovich et al. (2022a).

Building upon this basic pipeline, we develop our method, 3DGS-DET, by introducing two novel designs to improve the 3DGS representation, as illustrated in the bottom row of Fig. 2. These designs are detailed in the following sections Sec. 3.3 and Sec. 3.4.

### 3.3 BOUNDARY GUIDANCE

Given the fact that 3DGS reconstruction is derived from 2D images, we design the novel Boundary Guidance strategy by incorporating 2D Boundary Guidance to achieve a more suitable 3D spatial distribution of Gaussian blobs for detection. In this section, we present our Boundary Guidance strategy in detail. As illustrated in the top row of Fig. 3, to provide the guidance priors for 3DGS reconstruction, we first generate category-specific boundaries for posed images:

$$B_{\text{bd}} = \text{H}_{\text{bd}}(I) = \{b_{\text{bd}}^c\} \quad c \in C, \tag{8}$$

where $\text{H}_{\text{bd}}$ is the boundary generator, and $b_{\text{bd}}^c$ represents the binary boundary map for category $c$. If $b_{\text{bd}}^c(x,y) = 1$, the pixel at $(x,y)$ belongs to the boundary for objects of category $c$. The set $C$

includes all categories. In practice, the operations of $H_{bd}$ are as follows: we use Grounded SAM (Ren et al., 2024) to generate category-specific masks. Then, the Suzuki-Abe algorithm (Suzuki et al., 1985) is employed to extract the boundaries of these masks, along with category information. The category-specific boundaries are then overlaid on the posed images in different colors:

$$I_{bd}(x, y) = I(x, y) \cdot \left(1 - \sum_{c \in C} b_{bd}^c(x, y)\right) + \sum_{c \in C} b_{bd}^c(x, y) \cdot \text{color}(c), \tag{9}$$

where $I_{bd}(x, y)$ is the pixel at position $(x, y)$ of the final image with overlaid boundaries. $I(x, y)$ is from the original image, $b_{bd}^c(x, y)$ is the boundary map for category $c$, and $\text{color}(c)$ is the color associated with category $c$. These $I_{bd}$ images are used as ground truth to train the 3DGS representation $G_{bd}$ by the following loss:

$$L_{render} = (1 - \lambda)L_1(I, I_{bd}) + \lambda L_{D\text{-SSIM}}(I, I_{bd}). \tag{10}$$

To effectively reduce $L_{render}$ during training, it is crucial to ensure the rendering quality of boundaries and the multi-view stability of boundaries. In this way, the Boundary Guidance lead 3DGS to incorporate boundary prior information into the 3D space. As shown in Fig. 2 (better viewed when zoomed in), 3DGS trained with Boundary Guidance demonstrates improved spatial distribution of Gaussian blobs compared to those trained without it, without introducing additional learnable parameters.

### 3.4 BOX-FOCUSED SAMPLING

Considering that 2D images often include numerous background pixels, leading to densely reconstructed 3DGS with many noisy Gaussian blobs representing the background, negatively affecting detection. To reduce the excessive background blobs, in this section, we propose the Box-Focused Sampling strategy in detail. As depicted in the bottom row of Fig. 3, to provide priors for the following sampling, we utilize a 2D object detector to identify object bounding boxes:

$$B_{bb} = H_{bb}(I) = \{(b_{bb}, p^C)\}, \tag{11}$$

where $H_{bb}$ is the box detector, and we select Grounding DINO (Liu et al., 2023) as the detector in our experiments. Here, $b_{bb}$ denotes the bounding box positions, and $p^C$ is the probability vector for the box belonging to each category in $C$. We define $p_{max} = \max_{c \in C} p^c$ as the highest category probability for a given bounding box, which helps to establish object probability spaces in later step. Then, we project the 2D boxes into 3D space:

$$F_{ft} = \left\{ K^{-1} \begin{bmatrix} x_i \\ y_i \\ z \end{bmatrix} \mid (x_i, y_i) \in b_{bb}, z \in \{z_{min}, z_{max}\} \right\}, \tag{12}$$

where $F_{ft}$ is the projected 3D frustum from $b_{bb}$, and $K^{-1}$ is the inverse camera matrix used to map 2D bounding box corners $(x_i, y_i)$ and depth values $z_{min}$ and $z_{max}$ into 3D space. Next, we establish object probability spaces using $F_{ft}$ and $p_{max}$. Specifically, for each bounding box, the maximum probability $p_{max}$ models the likelihood of each Gaussian blob within the corresponding frustum being an object blob:

$$p_{obj}(g_i \mid g_i \in F_{ft}) = p_{max}, \tag{13}$$

where $p_{obj}(g_i \mid g_i \in F_{ft})$ indicates the probability of each Gaussian blob $g_i$ within frustum $F_{ft}$ being an object blob. To integrate priors from different view frustums, we select the maximum probability as the aggregated probability:

$$p_{agr}(g_i) = \max_{v \in V} p_{obj}(g_i \mid g_i \in F_{ft}^v), \tag{14}$$

where $p_{agr}(g_i)$ is the aggregated probability for Gaussian blob $g_i$, and $V$ represents the set of all views. Gaussian blobs not belonging to any frustum are assigned a small probability $p_{bg}$, set to 0.01 in practice. In this way, we obtain the object probability spaces $P_{obj}$, where each Gaussian blob has an associated probability of being an object. We then perform independent probabilistic sampling based on $P_{obj}$ to achieve Box-Focused Sampling, resulting in the sampled Gaussian set $\hat{G}_{bd}^{bs}$ as:

$$\hat{G}_{bd}^{bs} = \{g \mid g \sim P_{obj}(g)\}. \tag{15}$$

In this way, it allows object blobs to be better preserved due to their higher probabilities, while most background points, having lower probabilities, are effectively reduced. Then, based on $\hat{G}_{bd}^{bs}$, we proceed with the training of the detector, as formulated by Equ. 5-Equ. 7 as described in Sec. 3.2. As shown in Fig. 2, 3DGS sampled via Box-Focused Sampling retains more object blobs and reduces background noise.

Table 1: Comparison of mAP@0.25 across different methods on ScanNet. The first block includes methods using non-view-synthesis representations, such as point cloud, RGB-D, and multi-view images. The second block includes methods utilizing view-synthesis representations (NeRF-based and our 3DGS-based method). Our 3DGS-DET significantly outperforms the NeRF-based method NeRF-Det by 6.6 points. For other representations, 3DGS-DET surpasses all methods except for the point-cloud-based methods, FCAF3D and CAGroup3D, which have inherent advantages by directly using sensor-captured 3D data, specifically point clouds, as input.

| Methods | cab | bed | chair | sofa | tabl | door | wind | bkshf | pic | cntr |
|---|---|---|---|---|---|---|---|---|---|---|
| Seg-Cluster (Wang et al., 2018) | 11.8 | 13.5 | 18.9 | 14.6 | 13.8 | 11.1 | 11.5 | 11.7 | 0.0 | 13.7 |
| Mask R-CNN (He et al., 2017) | 15.7 | 15.4 | 16.4 | 16.2 | 14.9 | 12.5 | 11.6 | 11.8 | 19.5 | 13.7 |
| SGPN (Wang et al., 2018) | 20.7 | 31.5 | 31.6 | 40.6 | 31.9 | 16.6 | 15.3 | 13.6 | 0.0 | 17.4 |
| 3D-SIS (Hou et al., 2019) | 12.8 | 63.1 | 66.0 | 46.3 | 26.9 | 8.0 | 2.8 | 2.3 | 0.0 | 6.9 |
| 3D-SIS (w/ RGB) (Hou et al., 2019) | 19.8 | 69.7 | 66.2 | 71.8 | 36.1 | 30.6 | 10.9 | 27.3 | 0.0 | 10.0 |
| VoteNet (Qi et al., 2019) | 36.3 | 87.9 | 88.7 | 89.6 | 58.8 | 47.3 | 38.1 | 44.6 | 7.8 | 56.1 |
| FCAF3D (Rukhovich et al., 2022a) | 57.2 | 87.0 | 95.0 | 92.3 | 70.3 | 61.1 | 60.2 | 64.5 | 29.9 | 64.3 |
| CAGroup3D (Wang et al., 2022a) | 60.4 | 93.0 | 95.3 | 92.3 | 69.9 | 67.9 | 63.6 | 67.3 | 40.7 | 77.0 |
| ImGeoNet (Tu et al., 2023) | 40.6 | 84.1 | 74.8 | 75.6 | 59.9 | 40.4 | 24.7 | 60.1 | 4.2 | 41.2 |
| CN-RMA (Shen et al., 2024a) | 42.3 | 80.0 | 79.4 | 83.1 | 55.2 | 44.0 | 30.6 | 53.6 | 8.8 | 65.0 |
| ImVoxelNet (Rukhovich et al., 2022b) | 30.9 | 84.0 | 77.5 | 73.3 | 56.7 | 35.1 | 18.6 | 47.5 | 0.0 | 44.4 |
| NeRF-Det (Xu et al., 2023) | 37.6 | 84.9 | 76.2 | 76.7 | 57.5 | 36.4 | 17.8 | 47.0 | 2.5 | 49.2 |
| 3DGS-DET (Our basic pipeline) | 39.6 | 82.5 | 75.8 | 78.0 | 53.6 | 36.1 | 26.9 | 41.8 | 11.9 | 56.0 |
| 3DGS-DET (Our basic pipeline+BG) | 38.9 | 83.5 | 81.7 | 82.6 | 54.4 | 36.2 | 26.0 | 39.6 | 13.5 | 52.8 |
| 3DGS-DET (Our basic pipeline+BG+BS) | 44.1 | 82.7 | 81.7 | 79.6 | 56.0 | 35.4 | 27.6 | 45.2 | 17.3 | 61.9 |

| Methods | desk | curt | fridg | showr | toil | sink | bath | ofurn | mAP@0.25 |
|---|---|---|---|---|---|---|---|---|---|
| Seg-Cluster (Wang et al., 2018) | 12.2 | 12.4 | 11.2 | 18.0 | 19.5 | 18.9 | 16.4 | 12.2 | 13.4 |
| Mask R-CNN (He et al., 2017) | 14.4 | 14.7 | 21.6 | 18.5 | 25.0 | 24.5 | 24.5 | 16.9 | 17.1 |
| SGPN (Wang et al., 2018) | 14.1 | 22.2 | 0.0 | 0.0 | 72.9 | 52.4 | 0.0 | 18.6 | 22.2 |
| 3D-SIS (Hou et al., 2019) | 33.3 | 2.5 | 10.4 | 12.2 | 74.5 | 22.9 | 58.7 | 7.1 | 25.4 |
| 3D-SIS (w/ RGB) (Hou et al., 2019) | 46.9 | 14.1 | 53.8 | 36.0 | 87.6 | 43.0 | 84.3 | 16.2 | 40.2 |
| VoteNet (Qi et al., 2019) | 71.7 | 47.2 | 45.4 | 57.1 | 94.9 | 54.7 | 92.1 | 37.2 | 58.7 |
| FCAF3D (Rukhovich et al., 2022a) | 71.5 | 60.1 | 52.4 | 83.9 | 99.9 | 84.7 | 86.6 | 65.4 | 71.5 |
| CAGroup3D (Wang et al., 2022a) | 83.9 | 69.4 | 65.7 | 73.0 | 100.0 | 79.7 | 87.0 | 66.1 | 75.12 |
| ImGeoNet (Tu et al., 2023) | 70.9 | 33.7 | 54.4 | 47.5 | 95.2 | 57.5 | 81.5 | 36.1 | 54.6 |
| CN-RMA (Shen et al., 2024a) | 70.0 | 44.9 | 44.0 | 55.2 | 95.4 | 68.1 | 86.1 | 49.7 | 58.6 |
| ImVoxelNet (Rukhovich et al., 2022b) | 65.5 | 19.6 | 58.2 | 32.8 | 92.3 | 40.1 | 77.6 | 28.0 | 49.0 |
| NeRF-Det (Xu et al., 2023) | 52.0 | 29.2 | 68.2 | 49.3 | 97.1 | 57.6 | 83.6 | 35.9 | 53.3 |
| 3DGS-DET (Our basic pipeline) | 69.8 | 36.7 | 38.3 | 55.3 | 93.5 | 64.0 | 80.8 | 37.5 | 54.3 |
| 3DGS-DET (Our basic pipeline+BG) | 68.6 | 45.2 | 52.7 | 45.0 | 98.3 | 69.6 | 84.3 | 48.0 | 56.7 |
| 3DGS-DET (Our basic pipeline+BG+BS) | 72.8 | 40.7 | 56.6 | 71.9 | 98.5 | 72.2 | 88.3 | 46.7 | **59.9** (+6.6) |

## 4 EXPERIMENTS

### 4.1 EXPERIMENTAL SETUP

**Dataset:** To thoroughly evaluate the performance of our proposed method in 3D detection tasks, we selected two representative datasets: ScanNet (Dai et al., 2017) and ARKitScene (Baruch et al., 2021). ScanNet is a large-scale indoor scene dataset containing over 1,500 real-world 3D scanned scenes, encompassing various complex indoor environments such as residential spaces, offices, and classrooms. The ARKitScene dataset is constructed from RGB-D image sequences, offering detailed geometric information and precise object annotations. For each scene, a maximum of 600 posed images are extracted. The category settings follow the standard 18 categories for ScanNet and 17 categories for ARKitScene.

**Metrics:** We use mAP@0.25 and mAP@0.5 as the primary evaluation metrics. Mean Average Precision (mAP) is calculated at different IoU thresholds, providing a comprehensive measure of the detection model's performance across various categories.

**Implementation Details:** For training 3DGS, we follow Kerbl et al. (2023) to initialize the 3D coordinates of Gaussian blobs using Structure from Motion (SfM) points. The training hyperparameters are the same as those in Kerbl et al. (2023). We employ pretrained GroundedSAM (Ren et al., 2024) and the Suzuki-Abe algorithm (Suzuki et al., 1985) as the boundary detector in Boundary Guidance. The pretrained GroundingDINO (Liu et al., 2023) is used as the box detector in the Box-Focused Sampling strategy. For the detection tool, we utilize the FCAF3D (Rukhovich et al., 2022a) architecture implemented in MMDetection3D (Contributors, 2020). The training hyperparameters are the same as those in FCAF3D. In our ablation study, to ensure a fair comparison, all model versions are trained with the same hyperparameters, such as the same number of epochs, specifically 12 epochs. All the ablation experiments (Sec. 4.3) are conducted on ScanNet.

## 4.2 MAIN RESULTS

**Quantitative Results**. For the **ScanNet** dataset, we present the mAP@0.25 and mAP@0.5 performances of various methods in Tab. 1 of the main paper and Tab. 6 of the Appendix, respectively. Note that some methods did not report mAP@0.5 in previous studies, resulting in blank entries for these methods in Tab. 6 of the Appendix.

In both Tab. 1 and Tab. 6 of the Appendix, the methods listed in the first block (Wang et al., 2018; He et al., 2017; Hou et al., 2019; Qi et al., 2019; Rukhovich et al., 2022a; Wang et al., 2022a; Tu et al., 2023; Shen et al., 2024a; Rukhovich et al., 2022b) are non-view-synthesis representation-based 3D detection methods. These methods utilize point clouds, RGB-D data, or multi-view images for 3D object detection. The second block consists of view-synthesis representation-based 3DOD methods, including NeRF-Det (Hu et al., 2023) and our proposed 3DGS-DET. NeRF-Det is the closest work to ours, leveraging Neural Radiance Fields (NeRF). Our approach variants are detailed as follows: '3DGS-DET (Our basic pipeline)' represents the basic pipeline method established in Sec. 3.2. '3DGS-DET (Our basic pipeline+BG)' incorporates the proposed Boundary Guidance as detailed in Sec. 3.3. '3DGS-DET (Our basic pipeline+BG+BS)' is our full method, utilizing both Boundary Guidance and Box-Focused Sampling as described in Sec. 3.4. As illustrated in Tab. 1 and Tab. 6 of the Appendix, all versions of our methods significantly outperform NeRF-Det. Notably, our full method ('Our basic pipeline+BG+BS') surpasses the state-of-the-art NeRF-based method, NeRF-Det, by **+6.6** on mAP@0.25 and **+8.1** on mAP@0.5, showcasing the superiority of our approach.

Table 2: Comparison of the 'whole-scene' performance on the ARKITScenes validation set. Our 3DGS-DET significantly outperforms NeRF-Det by 31.5 points. Note that we follow the setup described in the NeRF-Det (Xu et al., 2023) supplementary materials: 'In our experiments, we utilize the subset of the dataset with low-resolution images', considering it is the closest work to ours. Other methods that do not use the same setting are not listed in this table.

| Methods | cab | fridg | shlf | stove | bed | sink | wshr | tolt | bthtb |
|---|---|---|---|---|---|---|---|---|---|
| ImVoxelNet (Rukhovich et al., 2022b) | 32.2 | 34.3 | 4.2 | 0.0 | 64.7 | 20.5 | 15.8 | 68.9 | 80.4 |
| NeRF-Det (Xu et al., 2023) | 36.1 | 40.7 | 4.9 | 0.0 | 69.3 | 24.4 | 17.3 | 75.1 | 84.6 |
| 3DGS-DET (Ours) | 45.2 | 84.4 | 33.3 | 41.4 | 87.3 | 75.5 | 67.6 | 87.2 | 90.8 |

| Methods | oven | dshwshr | frplce | stool | chr | tble | TV | sofa | **mAP@.25** |
|---|---|---|---|---|---|---|---|---|---|
| ImVoxelNet (Rukhovich et al., 2022b) | 9.9 | 4.1 | 10.2 | 0.4 | 5.2 | 11.6 | 3.1 | 35.6 | 23.6 |
| NeRF-Det (Xu et al., 2023) | 14.0 | 7.4 | 10.9 | 0.2 | 4.0 | 14.2 | 5.3 | 44.0 | 26.7 |
| 3DGS-DET (Ours) | 74.3 | 6.0 | 56.4 | 26.3 | 70.3 | 60.6 | 0.7 | 81.8 | **58.2** (+31.5) |

Regarding the **ARKitScene** dataset, considering NeRF-Det is the closest work to ours, we follow the same setup described in the NeRF-Det (Xu et al., 2023) supplementary materials: 'In our experiments, we utilize the subset of the dataset with low-resolution images.' Similarly, we adopt the same subset of the ARKitScenes dataset. Other methods that report performance on ARKitScene use the full dataset, so our 3DGS-DET is only compared with ImVoxelNet and NeRF-Det under the same conditions as described in NeRF-Det. The results in Tab. 2 demonstrate that 3DGS-DET performs better across most categories, achieving an mAP@0.25 of 58.2, which significantly outperforms NeRF-Det by **+31.5**, highlighting the superiority of our method.

**Qualitative results**. We provide a qualitative comparison with NeRF-Det in Fig. 4. As shown, our methods detect more objects in the scene with greater positional accuracy compared to NeRF-Det (Xu et al., 2023), demonstrating the superiority of our approach. More qualitative comparisons can be found in Fig. 6 and Fig. 7 in the Appendix.

## 4.3 ABLATION STUDY

### 4.3.1 ANALYSIS ON THE EFFECT OF PROPOSED DESIGNS

In this section, we demonstrate the effectiveness of our contributions by first presenting the performance of our proposed basic 3DGS detection pipeline and then incrementally incorporating our additional designs to analyze the resulting performance improvements.

**Our Proposed Basic 3DGS Detection Pipeline**. As shown in Tab. 1, '3DGS-DET (Our basic pipeline)' represents our proposed detection pipeline utilizing 3DGS, as described in Section 3.2. Benefiting from the advantages of 3DGS as an explicit scene representation, our basic pipeline surpasses NeRF-Det by 1 point (54.3 vs. 53.3), underscoring the significance of introducing 3DGS into 3DOD for the first time.

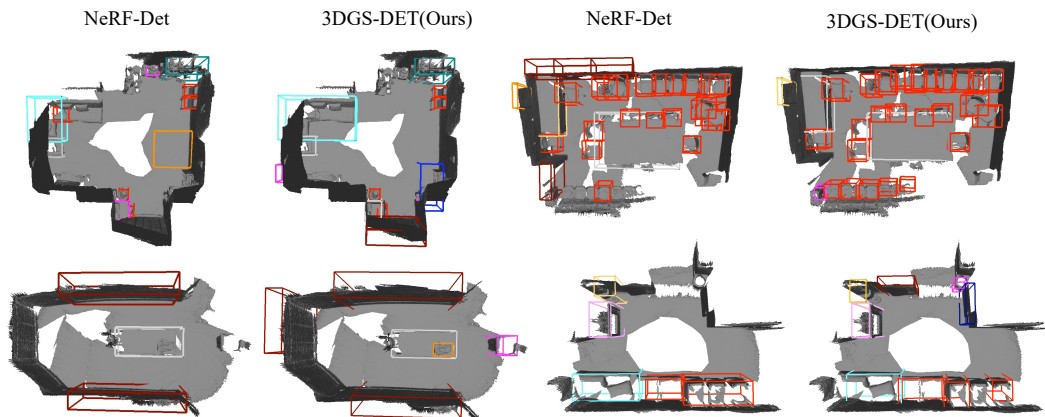

Figure 4: Qualitative comparison. Our methods identify more 3D objects in the scene with better positional precision, highlighting the advantages of our approach over NeRF-Det (Xu et al., 2023). In this figure, the scene is represented using mesh to clearly show the boxes.

**Boundary Guidance**. '3DGS-DET (Our basic pipeline+BG)' incorporates the proposed Boundary Guidance as detailed in Sec. 3.3. Introducing Boundary Guidance into the basic pipeline results in a significant improvement of 2.4 points (56.7 vs. 54.3), demonstrating the effectiveness of the proposed Boundary Guidance. To further explore the impact of Boundary Guidance on 3DGS representations, we present a visual comparison of the spatial distribution of trained Gaussian blobs in Fig. 8 in the Appendix. As we can see, Gaussian blobs trained with Boundary Guidance demonstrate clearer spatial distribution and more distinct differentiation between objects and the background. We also present rendered images from different views by 3DGS trained with Boundary Guidance in Fig. 9 and Fig. 10 in the Appendix. As can be observed, the category-specific boundaries are clearly rendered and show multi-view stability, indicating that the 3D representation has effectively embedded the priors from Boundary Guidance. All these results clearly verify the effectiveness of the proposed Boundary Guidance for 3D detection with 3DGS.

**Box-Focused Sampling**. Furthermore, we introduce Box-Focused Sampling detailed in Sec. 3.4, represented by '3DGS-DET (Our basic pipeline+BG+BS)' in Tab. 1. This addition leads to a further performance boost of 3.2 points (59.9 vs. 56.7), proving the effectiveness of Box-Focused Sampling. The visual comparison of sampled Gaussian blobs is shown in Fig. 11 in the Appendix. We can observe that the proposed Box-Focused Sampling significantly retains more object blobs and suppresses noisy background blobs.

Table 3: Ablation study on guidance from different priors.

| Different Priors | mAP@0.25 | mAP@0.5 |
|---|---|---|
| 2D Center Point | 54.4 | 33.9 |
| 2D Mask | 54.9 | 34.2 |
| 2D Boundary (ours) | **56.7** | **36.9** |

Table 4: Ablation study on different sampling methods.

| Sampling Methods | mAP@0.25 | mAP@0.5 |
|---|---|---|
| Random Sampling | 56.7 | 36.9 |
| Farthest Point Sampling | 57.4 | 37.6 |
| Box-focused Sampling (ours) | **59.9** | **37.8** |

### 4.3.2 ABLATION STUDY ON GUIDANCE FROM DIFFERENT PRIORS

In this section, we analyze the impact of guidance from various priors. As described in Sec. 3.3, we utilize the object's boundary as the guidance prior. Here, we perform an ablation study considering the object's center point and mask as alternative priors. To obtain the center point, we detect the object's bounding box using GroundingDINO (Liu et al., 2023) and compute its center coordinates. The mask is generated with GroundedSAM (Ren et al., 2024). Note that all priors are category-specific, with each class associated with a fixed color. These priors are overlaid on the posed images, as shown in Fig. 5, and then used to train the 3DGS for detection. Tab. 3 presents the detection performance for 3DGS trained with the different priors. As reported in Tab. 3, the 3DGS-DET method using boundary guidance achieved 56.7% in mAP@0.25 and 36.9% in mAP@0.5, demonstrating significant superiority over the center point and mask priors.

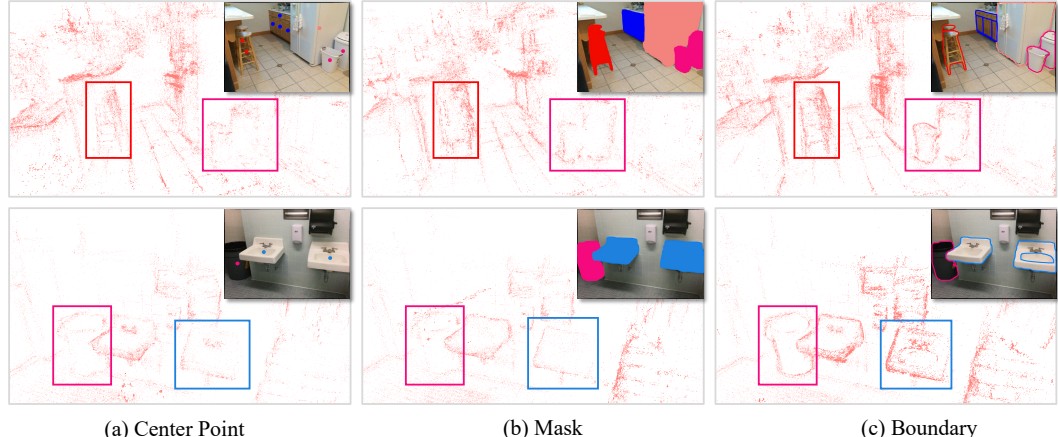

| (a) Center Point | (b) Mask | (c) Boundary |

Figure 5: Analysis of guidance from different priors: (a) Center Point Guidance, (b) Mask Guidance, and (c) Boundary Guidance. In (a) and (b), the spatial distribution of Gaussian blobs for objects like the chair, trash bin and sink is incomplete and ambiguous. Gaussian blobs trained with Boundary Guidance exhibit a clearer spatial distribution. The reason behind this phenomenon is that the center point provides only positional guidance, lacking richer information like shape or size. The mask highlights shape and size but hides the object's surface, reducing texture and geometric information. Boundary Guidance offers positional cues and richer information, such as shape and size, while preserving texture and geometric details on the object's surface, leading to the best performance.

Let's explore the visualizations for further insights. In (a) and (c) of Fig. 5, we observe that the spatial distribution of Gaussian blobs with Point Guidance is less distinct compared to Boundary Guidance. This is because the center point provides only positional guidance, lacking richer information like shape or size, making it less effective compared to the boundary prior. For the mask prior, as shown in (b) and (c) of Fig. 5, the Gaussian blobs' spatial distribution with Mask Guidance is more ambiguous than with the Boundary Guidance. Although the mask highlights shape and size information, it hides the object's surface, reducing texture and geometric information, thus being less effective than the boundary prior. Overall, Boundary Guidance offers positional cues and richer information such as shape and size while preserving texture and geometric details on the object's surface, leading to the best performance.

### 4.3.3 ANALYSIS ON DIFFERENT SAMPLING METHODS

In this section, we compare two additional sampling methods with our Box-Focused Sampling: 1) Random Sampling and 2) Farthest Point Sampling (Qi et al., 2017b). The latter iteratively selects points farthest from those already chosen, ensuring even distribution for better scene coverage, focusing on global distribution rather than specific geometric features of objects. The results in Tab. 4 demonstrate that our Box-Focused Sampling achieves the highest performance, with mAP@0.25 and mAP@0.5 reaching 59.9% and 37.8%, respectively. This is because 3DGS often contain excessive background blobs. Our Box-Focused Sampling is specifically designed to preserve more object-related blobs while suppressing noisy background blobs. In contrast, other sampling methods primarily focus on global scenes without differentiation between objects and background blobs.

## 5 CONCLUSION

In this work, we introduce 3D Gaussian Splatting (3DGS) into 3D Object Detection (3DOD) for the first time. We propose 3DGS-DET, a novel approach that leverages Boundary Guidance and Box-Focused Sampling to enhance 3DGS for 3DOD. Our method effectively addresses the inherent challenges of 3DGS in 3D object detection by improving spatial distribution and reducing background noise. By incorporating 2D Boundary Guidance, we achieve clearer differentiation between objects and background, while Box-Focused Sampling retains more object points and minimizes background noise. Our method demonstrates significant improvements, with gains of +5.6 on mAP@0.25 and +3.7 on mAP@0.5 over the basic pipeline. It also outperforms state-of-the-art NeRF-based methods, achieving +6.6 on mAP@0.25 and +8.1 on mAP@0.5 on the ScanNet dataset, and an impressive +31.5 on mAP@0.25 on the ARKITScenes dataset. These results underscore the effectiveness and superiority of our designs.

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

# A APPENDIX

## A.1 PERFORMANCE ON NERF-RPN SETTING

Table 5: Performance on the NeRF-RPN setting, which targets class-agnostic box detection. Our method significantly outperforms NeRF-RPN in this setting.

| Methods | mAP@0.25 | mAP@0.5 |
|---|---|---|
| NeRF-RPN (Hu et al., 2023) | 55.5 | 18.4 |
| 3DGS-DET (ours) | **75.6** (+20.1) | **52.3** (+33.9) |

In this section, we adapt our 3DGS-DET to the NeRF-RPN Setting (Hu et al., 2023), which targets class-agnostic box detection. To achieve this, we labeled all the ground-truth boxes with a single 'object' category and trained 3DGS-DET accordingly. Additionally, NeRF-RPN uses a different train/validation split compared to the official ScanNet dataset, with its validation set overlapping the official ScanNet training set. To address this, we excluded the overlapping parts between the NeRF-RPN test set and the ScanNet official training set from our training data. We then used the remaining scenes for training, and tested on the same validation set provided by NeRF-RPN. As shown in Tab. 5, 3DGS-DET achieved an mAP@0.25 of 75.6% and an mAP@0.5 of 52.3%, significantly outperforming NeRF-RPN (Hu et al., 2023)'s 55.5% and 18.4%. This demonstrates the significant superiority of our method in the class-agnostic setting.

## A.2 FUTURE WORK

As the first work to introduce 3DGS into 3DOD, our paper mainly focuses on the primary stage of this pipeline: empowering 3DGS for 3DOD. Diverse experiments demonstrate that our designs can lead to significant improvements. Beyond empowering the 3DGS representation, a subsequent detector specifically designed for 3DGS could hold promise in the future. Besides, exploring joint training of 3DGS and the detector is also an interesting direction. We hope our exploration knowledge, open-source codes and data will inspire further research.

Table 6: Comparison of mAP@0.5 across different methods on ScanNet. The first block presents methods that employ non-view-synthesis representations, including point clouds, RGB-D, and multi-view images. The second block lists methods using view-synthesis representations, such as NeRF-based and our 3DGS-based techniques. Our 3DGS-DET significantly surpasses the NeRF-based NeRF-Det by 8.1 points. Among other representations, 3DGS-DET outperforms all except the point-cloud-based methods FCAF3D and CAGroup3D, which benefit from directly using sensor-captured 3D data, specifically point clouds, as input. Note that some methods did not report mAP@0.5 in previous works, resulting in blank entries for these methods.

| Methods | cab | bed | chair | sofa | tabl | door | wind | bkshf | pic | cntr |
|---|---|---|---|---|---|---|---|---|---|---|
| Seg-Cluster (Wang et al., 2018) | - | - | - | - | - | - | - | - | - | - |
| Mask R-CNN (He et al., 2017) | - | - | - | - | - | - | - | - | - | - |
| SGPN (Wang et al., 2018) | - | - | - | - | - | - | - | - | - | - |
| 3D-SIS (Hou et al., 2019) | 5.1 | 42.2 | 50.1 | 31.8 | 15.1 | 1.4 | 0.0 | 1.4 | 0.0 | 0.0 |
| 3D-SIS (w/ RGB) (Hou et al., 2019) | 5.7 | 50.3 | 52.6 | 55.4 | 22.0 | 10.9 | 0.0 | 13.2 | 0.0 | 0.0 |
| VoteNet (Qi et al., 2019) | 8.1 | 76.1 | 67.2 | 68.8 | 42.4 | 15.3 | 6.4 | 28.0 | 1.3 | 9.5 |
| FCAF3D (Rukhovich et al., 2022a) | 35.8 | 81.5 | 89.8 | 85.0 | 62.0 | 44.1 | 30.7 | 58.4 | 17.9 | 31.3 |
| CAGroup3D (Wang et al., 2022a) | 41.4 | 82.8 | 90.8 | 85.6 | 64.9 | 54.3 | 37.3 | 64.1 | 31.4 | 41.1 |
| ImGeoNet (Tu et al., 2023) | 15.8 | 74.8 | 46.5 | 45.7 | 39.9 | 8.0 | 2.9 | 32.9 | 0.3 | 7.9 |
| CN-RMA (Shen et al., 2024a) | 21.3 | 69.2 | 52.4 | 63.5 | 42.9 | 11.1 | 6.5 | 40.0 | 1.2 | 24.9 |
| ImVoxelNet (Rukhovich et al., 2022b) | 8.9 | 67.1 | 35.0 | 33.1 | 30.5 | 4.9 | 1.3 | 7.0 | 0.1 | 0.9 |
| NeRF-Det (Xu et al., 2023) | 12.0 | 68.4 | 47.8 | 58.3 | 42.8 | 7.1 | 3.0 | 31.3 | 1.6 | 11.6 |
| 3DGS-DET (Our basic pipeline) | 18.5 | 73.5 | 44.6 | 61.9 | 42.2 | 9.3 | 5.6 | 28.7 | 2.3 | 2.0 |
| 3DGS-DET (Our basic pipeline+BG) | 16.1 | 77.0 | 51.6 | 62.4 | 44.7 | 11.7 | 11.3 | 24.4 | 1.7 | 19.0 |
| 3DGS-DET (Our basic pipeline+BG+BS) | 19.2 | 73.8 | 52.7 | 65.2 | 46.2 | 9.6 | 8.2 | 31.8 | 4.2 | 20.9 |

| Methods | desk | curt | fridg | showr | toil | sink | bath | ofurn | mAP@0.5 |
|---|---|---|---|---|---|---|---|---|---|
| Seg-Cluster (Wang et al., 2018) | - | - | - | - | - | - | - | - | - |
| Mask R-CNN (He et al., 2017) | - | - | - | - | - | - | - | - | - |
| SGPN (Wang et al., 2018) | - | - | - | - | - | - | - | - | - |
| 3D-SIS (Hou et al., 2019) | 13.7 | 0.0 | 2.7 | 3.0 | 56.8 | 8.7 | 28.5 | 2.6 | 14.6 |
| 3D-SIS (w/ RGB) (Hou et al., 2019) | 23.6 | 2.6 | 24.5 | 0.8 | 71.8 | 8.9 | 56.4 | 6.9 | 22.5 |
| VoteNet (Qi et al., 2019) | 37.5 | 11.6 | 27.8 | 10.0 | 86.5 | 16.8 | 78.9 | 11.7 | 33.5 |
| FCAF3D (Rukhovich et al., 2022a) | 53.4 | 44.2 | 46.8 | 64.2 | 91.6 | 52.6 | 84.5 | 57.1 | 57.3 |
| CAGroup3D (Wang et al., 2022a) | 63.6 | 44.4 | 57.0 | 49.3 | 98.2 | 55.4 | 82.4 | 58.8 | 61.3 |
| ImGeoNet (Tu et al., 2023) | 43.9 | 4.3 | 24.0 | 2.0 | 68.8 | 24.5 | 61.7 | 17.4 | 28.9 |
| CN-RMA (Shen et al., 2024a) | 51.4 | 19.6 | 33.0 | 6.6 | 73.3 | 36.1 | 76.4 | 31.5 | 36.8 |
| ImVoxelNet (Rukhovich et al., 2022b) | 35.5 | 0.6 | 22.1 | 4.5 | 67.7 | 18.9 | 60.2 | 10.1 | 22.7 |
| NeRF-Det (Xu et al., 2023) | 46.0 | 5.8 | 26.0 | 1.6 | 69.0 | 25.5 | 55.8 | 21.1 | 29.7 |
| 3DGS-DET (Our basic pipeline) | 53.5 | 18.1 | 30.7 | 3.4 | 77.0 | 29.0 | 68.3 | 24.2 | 34.1 |
| 3DGS-DET (Our basic pipeline+BG) | 47.4 | 27.2 | 30.4 | 8.3 | 87.0 | 36.3 | 78.3 | 28.8 | 36.9 |
| 3DGS-DET (Our basic pipeline+BG+BS) | 52.4 | 22.2 | 36.9 | 15.7 | 82.6 | 35.1 | 74.0 | 28.9 | **37.8** (+8.1) |

NeRF-Det                                      3DGS-DET(Ours)

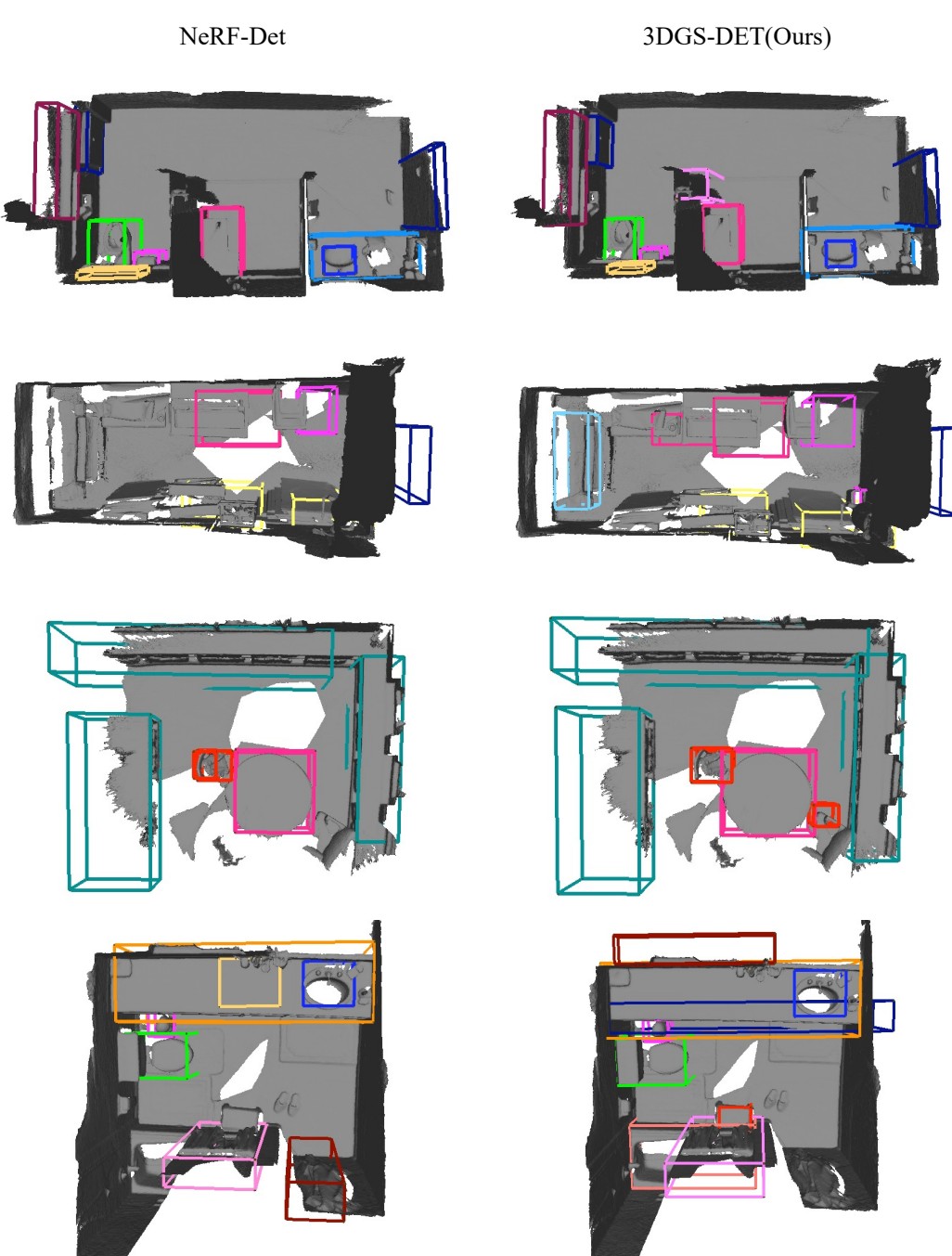

Figure 6: More qualitative comparison. Our methods identify more objects in the scene with better positional precision, highlighting the advantages of our approach over NeRF-Det (Xu et al., 2023). In this figure, the scene is represented using mesh to clearly display the boxes. Note that, Black and white boxes indicate predictions with incorrect categories, while boxes of other colors represent predictions with the correct category.

NeRF-Det                                3DGS-DET(Ours)

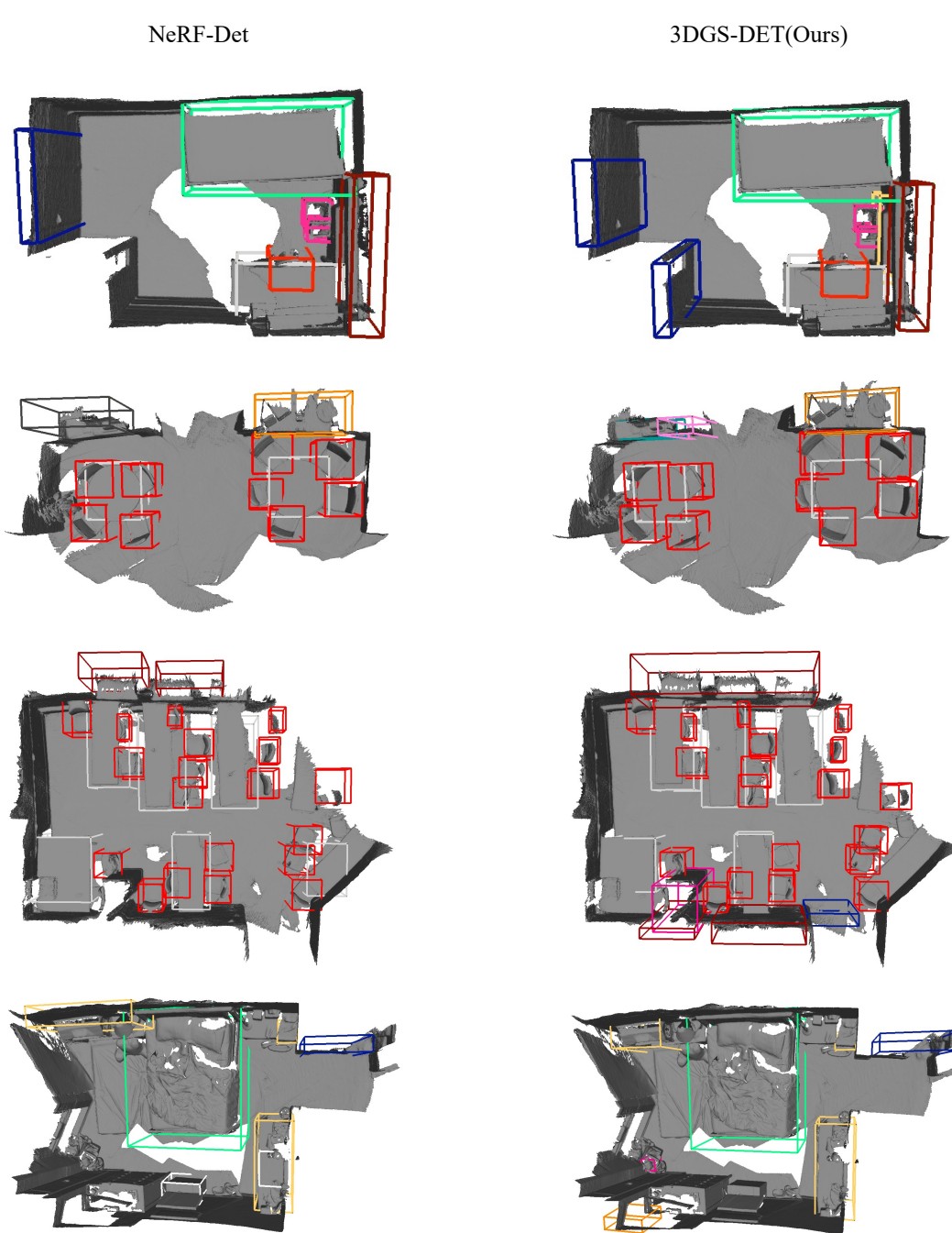

Figure 7: More qualitative comparison. Our methods identify more objects in the scene with better positional precision, highlighting the advantages of our approach over NeRF-Det (Xu et al., 2023). In this figure, the scene is represented using mesh to clearly display the boxes. Note that, Black and white boxes indicate predictions with incorrect categories, while boxes of other colors represent predictions with the correct category.

3DGS w/o Boundary Guidance          3DGS w/ Boundary Guidance

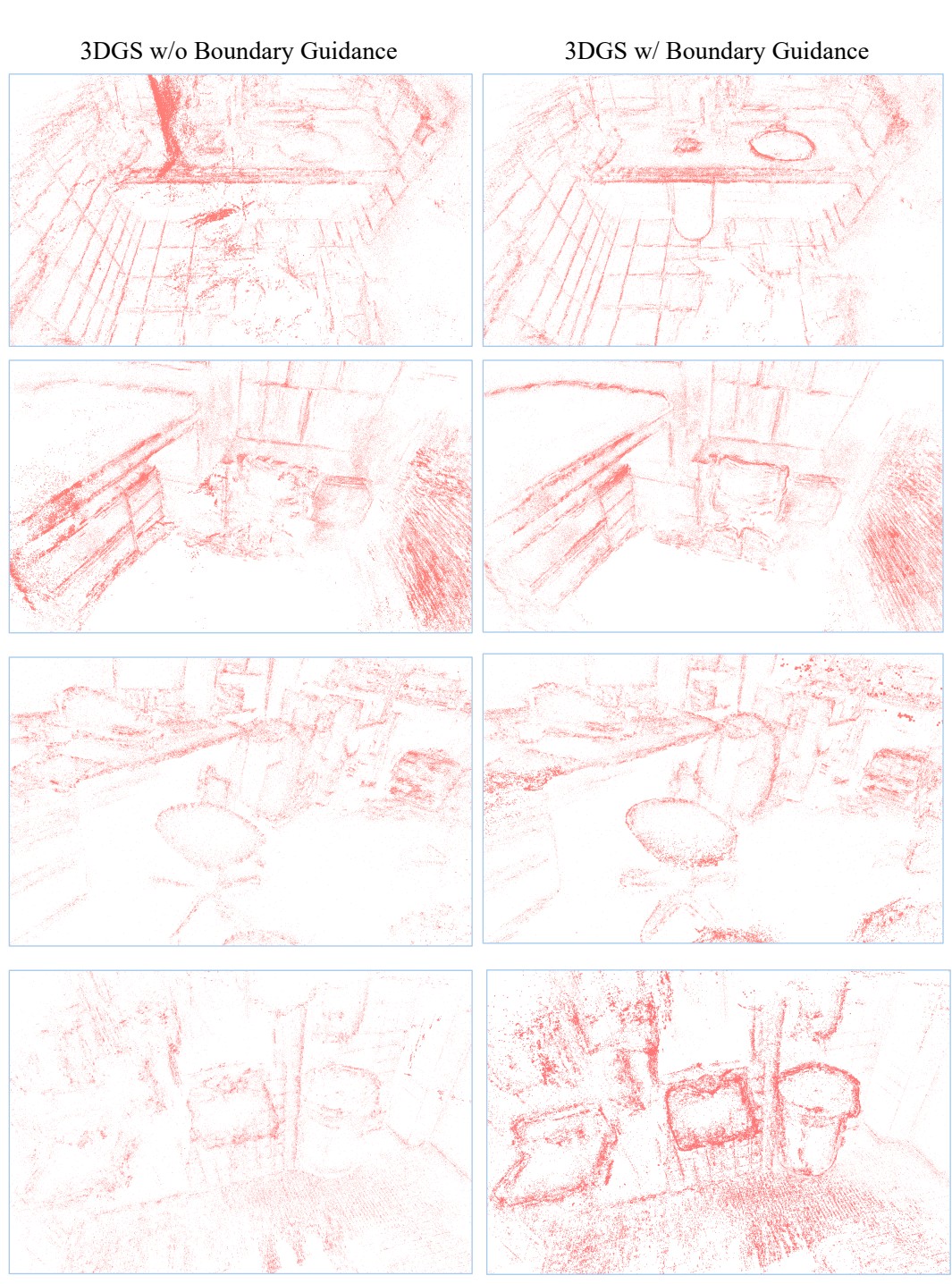

Figure 8: Analysis on the effect of Boundary Guidance. Gaussian blobs trained with Boundary Guidance exhibit clearer spatial distribution and more distinct differentiation between objects and background. Note that we visualize only the positions of the Gaussian blobs to highlight their spatial distribution, omitting other attributes.

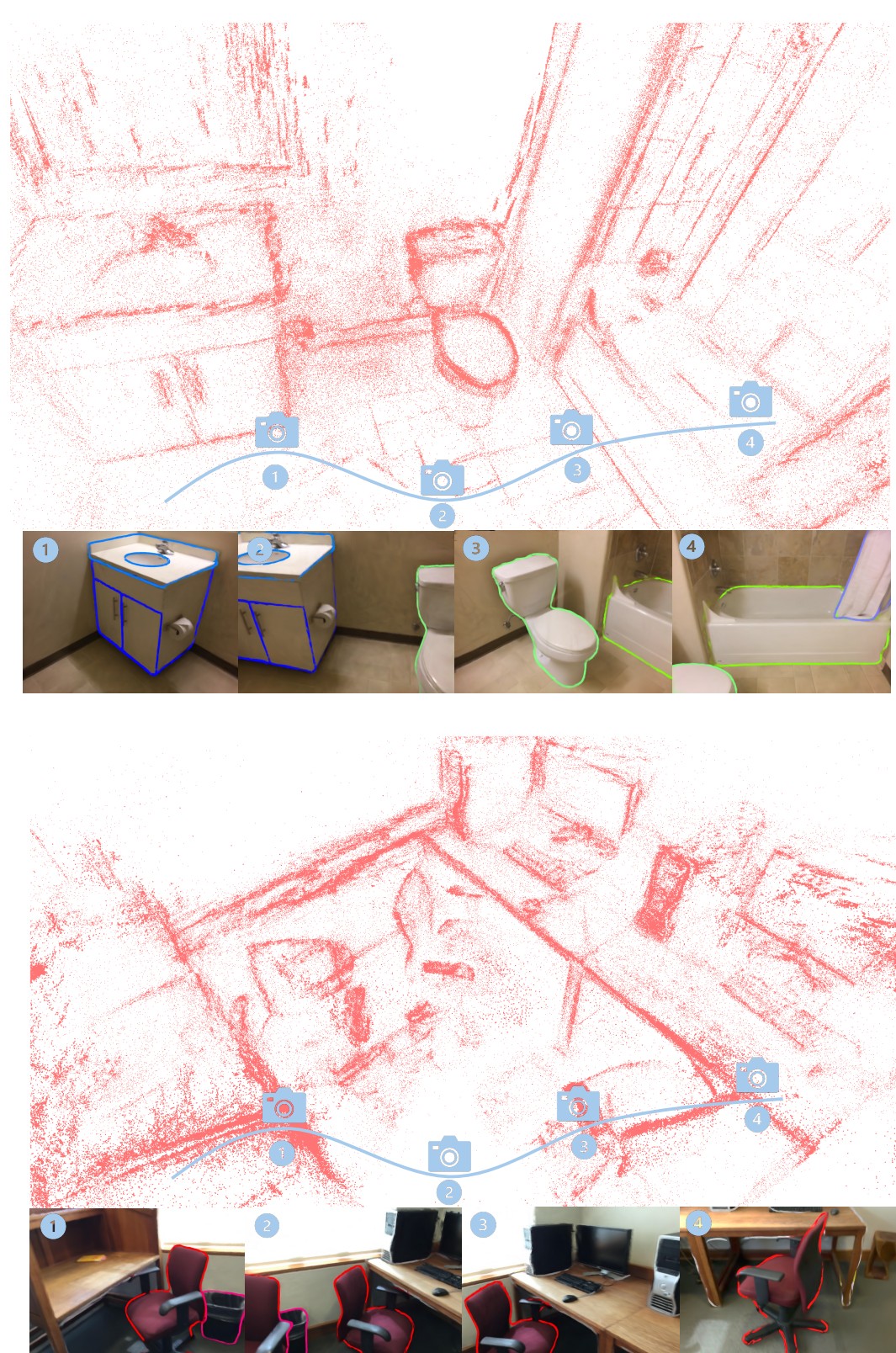

Figure 9: Rendered images from different views by 3DGS trained with Boundary Guidance. The category-specific boundaries are well rendered and exhibit multi-view stability, demonstrating that the 3D representation has successfully embedded the priors provided by Boundary Guidance.

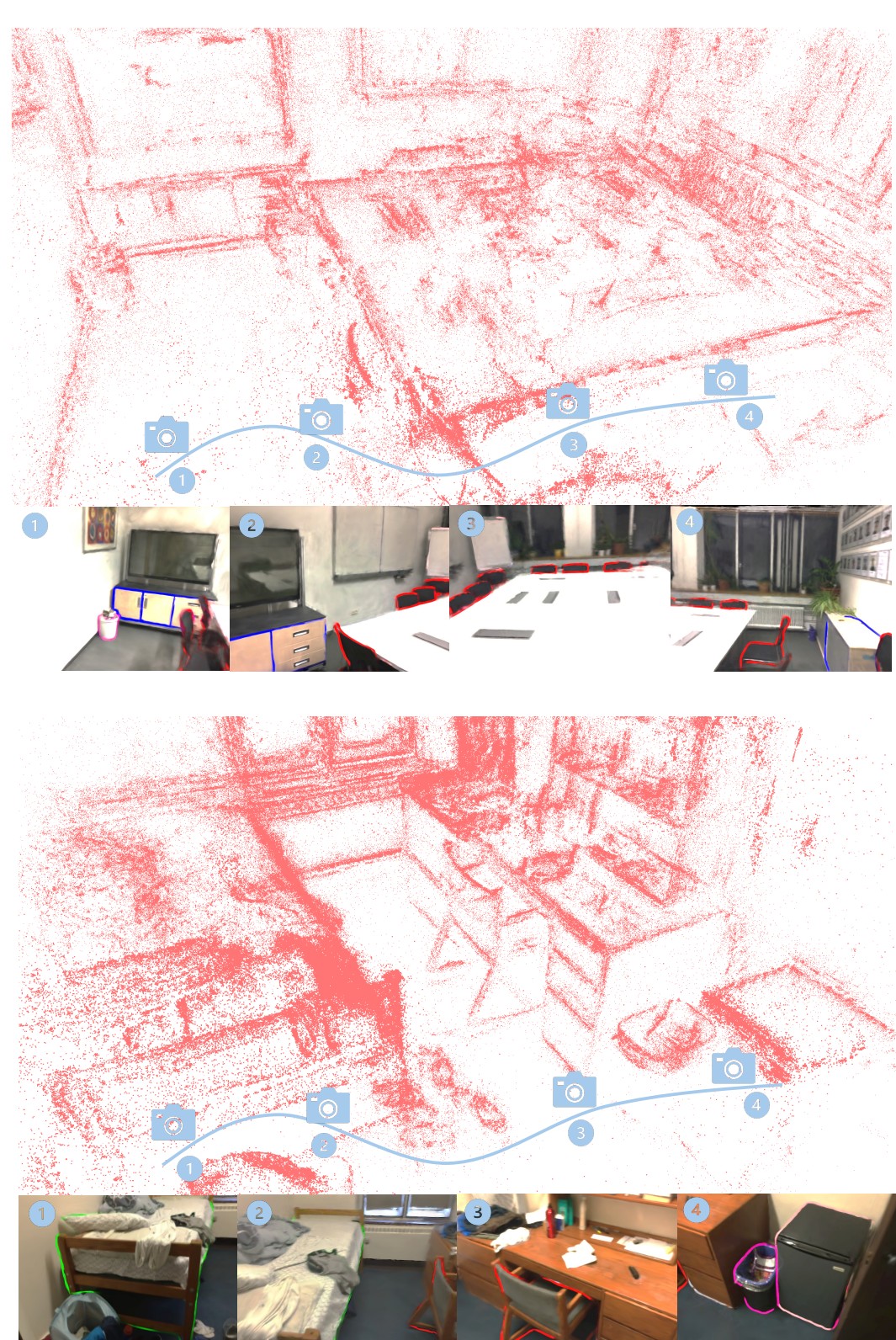

Figure 10: Rendered images from different views by 3DGS trained with Boundary Guidance. The category-specific boundaries are well rendered and exhibit multi-view stability, demonstrating that the 3D representation has successfully embedded the priors provided by Boundary Guidance.

Random Sampling         Box-Focused Sampling

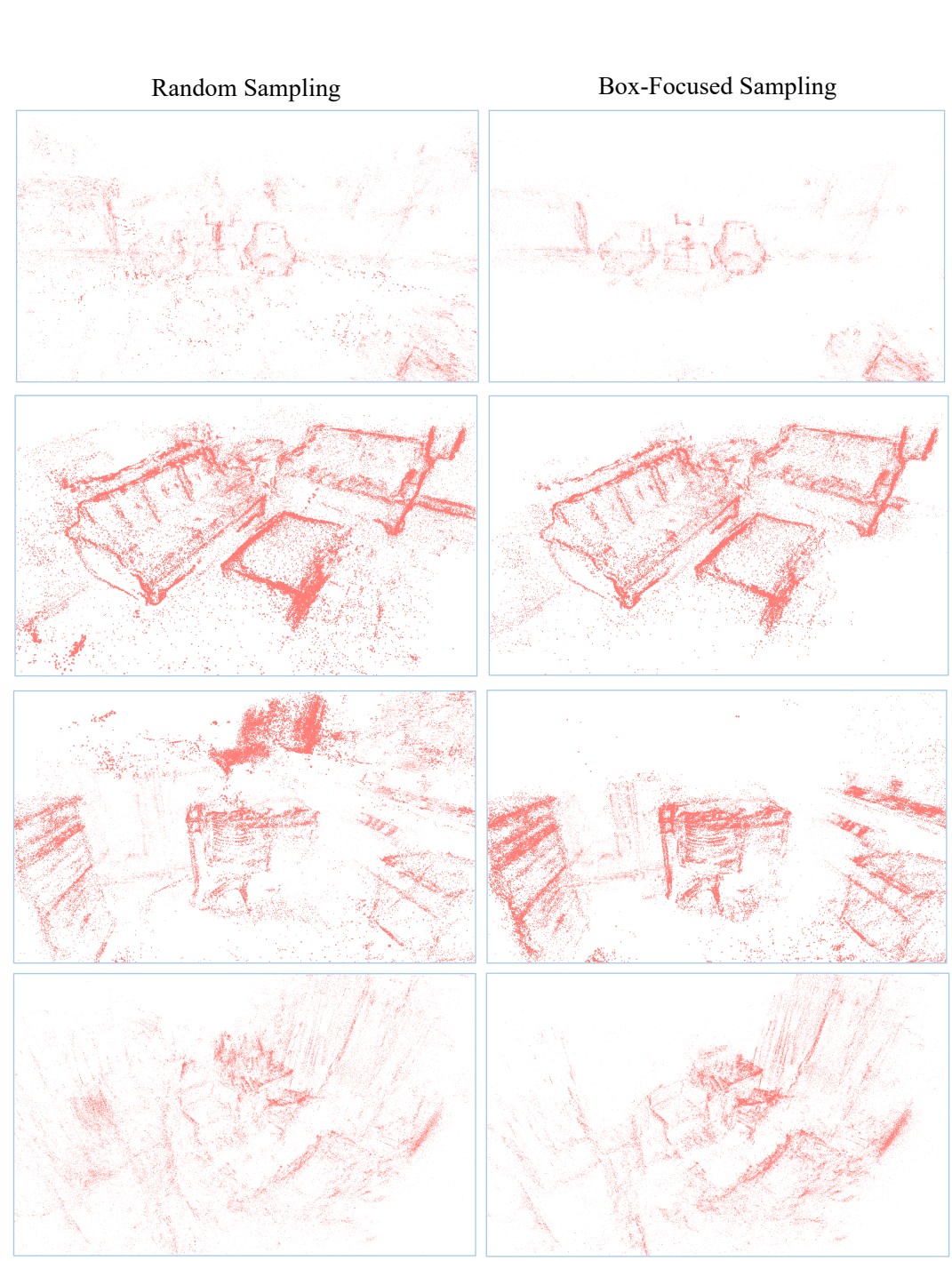

Figure 11: Analysis on the effect of Box-Focused Sampling. Box-Focused Sampling significantly retains more object blobs and reduces noisy background blobs. Note that we visualize only the positions of the Gaussian blobs to highlight their spatial distribution, omitting other attributes.

