# OpenReview forum: "3DGS-Det: Empower 3D Gaussian Splatting with Boundary Guidance and Box-Focused Sampling for 3D Object Detection"
_ICLR.cc/2025/Conference — ICLR 2025 Conference Withdrawn Submission_

### Official Review · Reviewer_XFhh · 2024-11-01

**Soundness:** 2
**Presentation:** 2
**Contribution:** 2
**Rating:** 5
**Confidence:** 3

**Summary:**

This paper pioneers using 3DGS for 3DOD, tackling two main challenges: (i) unclear spatial distribution of Gaussians, which affects object-background separation, and (ii) excessive background noise. They propose the 2D boundary guidance to improve spatial clarity and a Box-Focused sampling strategy for efficient object-focused sampling. The experiments show the improvement of object detection compared to their baseline and nerf-based method.

**Strengths:**

1. This paper solves 3DOD problem on 3DGS at the first time.
2. The improvement over NeRF-based detection is significant.

**Weaknesses:**

1. Combining neural rendering and 3D object detection is a relatively new direction. In the introduction, I think the authors should discuss more about the significance or application of performing 3D object detection with novel view synthesis, especially for the paper which first uses 3D Gaussians for object detection. In NeRF-based methods(NeRF-Det), their neural rendering and object detection mutually enhance each other; accurate geometry improves object detection, while the object detection task promotes geometric learning, ultimately enhancing the quality of both object detection and neural rendering. However, in this paper, the experiments only analyze 3D object detection, without evaluating the quality of rendering. Therefore, it is not possible to thoroughly investigate the impact between the rendering of 3D Gaussians and the object detection tasks under this method. In conclusion, why not independently perform the tasks of rendering and 3D object detection, if both yield better results when performed independently? Besides, for NeRF-based 3DOD, they train a feed-forward network for generalizable neural rendering, which could be benificial for perception. But this paper still need per-scene optimization (to my understanding).

2. The paper mentions that performing 3D object detection on 3DGS has higher rendering speed, but it does not compare the differences with NeRF-based methods in terms of training time, rendering time, and detection time.

**Questions:**

1. The paper mentions that background Gaussians affect detection, but there are backgrounds and target objects in point clouds as well. Do the point cloud-based 3DOD face the same issue? If not, why not directly perform point cloud object detection and rendering tasks separately?

2. From my understanding, this paper enhances the accuracy of 3D object detection by unprojecting 2D priors, i.e. edges and 2D detection information, into 3D space based on 3DGS. This enhancement seems reasonable. However, if this process only improves 3D object detection while not enhancing or even degrading neural rendering, it can be seen as a strategy solely for improving 3D object detection. The use of 3D Gaussians might not be necessary. For instance, could converting 3D Gaussians into point clouds for 3D object detection after training achieves similar improvements?

---

### Official Review · Reviewer_drQV · 2024-11-01

**Soundness:** 3
**Presentation:** 3
**Contribution:** 2
**Rating:** 5
**Confidence:** 4

**Summary:**

The paper proposes a guidance approach for 3DGS that facilitates the separation of 3D Gaussians representing distinct 3D object instances. Additionally, it introduces guided downsampling of the generated Gaussians, ensuring that most are associated with object instances rather than the background. The method shows improvements in two frequently used datasets.

**Strengths:**

- The paper proposes the first method that directly considers 3D Gaussians for 3D object detection.
- The method achieves good accuracy on two public datasets.

**Weaknesses:**

I have mixed feelings about the paper. On the one hand, the contributions appear questionable or minor. On the other, the method achieves competitive accuracy compared to baselines, nearly matching methods that do not rely on Gaussians. My primary concerns are as follows:

- The boundary guidance lacks justification. In Eq. 9, the authors incorporate a region constraint by linearly mixing the RGB color and a region-specific color at each pixel. However, they do not clarify how this region color is chosen — a critical detail, as the choice of color heavily impacts the optimization. For example, if the chosen region color is similar to nearby content (e.g., a white region color on a brown table near a white wall), the boundary constraint would fail. Additionally, the region's color likely interferes with both the object and surrounding colors, ultimately altering the Gaussians' colors, which may no longer represent the original scene. Maybe I misunderstood, but this choice seems odd; an alternative could have been to increase the number of channels per pixel or use the approach of LangSplat [a] to get the guidance.
- Section 3.4 describes Gaussian subsampling to retain the most representative objects. However, it is unclear how this impacts final accuracy. This approach seemingly does not enhance object detection but merely reduces the Gaussian cloud density. While this could affect the metrics, the practical value remains uncertain. Retaining a single well-localized Gaussian per object might optimize segmentation accuracy but would likely impair novel view synthesis and geometric accuracy. This raises the following point:
- It would be essential to assess novel view synthesis and geometric accuracy. Does this approach compromise other objectives of 3DGS, or do they remain intact?

Minor points and typos:
- Section 3.4: The process here is somewhat unclear. To my understanding, Gaussian splatting with boundary guidance runs first, followed by Gaussian subsampling to retain those likely corresponding to objects. This appears to involve guided sampling based on unspecified probabilities. The precise calculation of these "probability" scores is not explained. Regardless of my interpretation, the authors should clarify this part, as much of it is speculative.
- L080/L097: "empower" → "improve"
- L085: "distribution that is more differentiable" > I don't understand what the authors want to say here.
- L087: "to establish 3D object probability spaces" > "object probability spaces" is misleading. This term refers to a 3D subspace where objects are located, and it is unrelated to probabilities. The authors use "probability" inconsistently, such as in Eq. 14, to justify heuristics. Heuristics are acceptable, but mislabeling them as probabilities is not.
- L155: "significantly enhances the spatial distribution"—the opposite is true; it restricts distribution to independent object representations, losing background information.
- L316: What is "independent probabilistic sampling"?
- Fig. 2 does not effectively illustrate the pipeline. It should be improved.

[a] Qin, M., Li, W., Zhou, J., Wang, H. and Pfister, H., 2024. Langsplat: 3d language gaussian splatting. In Proceedings of the IEEE/CVF Conference on Computer Vision and Pattern Recognition (pp. 20051-20060).

**Questions:**

The authors can find the most important questions under the Weaknesses section.

---

### Official Review · Reviewer_VGvD · 2024-11-03

**Soundness:** 2
**Presentation:** 3
**Contribution:** 3
**Rating:** 5
**Confidence:** 4

**Summary:**

The manuscript introduces an approach to 3D object detection based on Gaussian Splats (GS) reconstructions. The method uses two mechanisms to guide the 3D GS reconstruction towards reconstructing the objects in the scene with high boundary fidelity before using a standard sparse 3D convolution-based 3D object detector on the GS parameters. The GS reconstructions are guided by (1) coloring object boundaries in the images and (2) resampling Gaussians with higher probability if they fall within an object frustum. The object boundary coloring leads to strong edges in the images which via the GS optimization translate into more Gaussians on object boundaries. The frustum resampling down-weighs background surfaces and focuses Gaussians to reconstruct primarily the objects in the scene. As a result the 3D detector has an easier job and performs well.

**Strengths:**

- The guidance of the GS reconstruction via boundary coloring and object frustum weighted resampling is clever and effective as can be observed qualitatively in the figures in the paper. These also quantitatively lead to improved 3D object detection as shown via ablation studies. These guidances in effect make condition the GS reconstruction on available 2D object evidence which is a very interesting paradigm. In the extreme one would aim to only reconstruct the objects?
- The improvement of 3D detection performance on ARKitScenes relative to NeRF-Det is impressive. I am quite curious to see some qualitative examples showing the improvements. For example the chair class jumped from 4% to 70.3%. I would love to see a qualitative comparison on a scene with many chairs.
- The paper is well written and has good figures to support the claims (effects of the two guidances on GS centers). As well as good qualitative figures showing the 3D bounding box detections (although I would really like to see some on ARKitScenes - see weaknesses).

**Weaknesses:**

- The ablation studies are effective but could be improved by adding lower bounds (no guidance mAP to Table 3) and upper bounds (sampling GS according to GT OBBs to table 4). The center-point guidance seems unnecessary - clearly a pixel-level optimization algorithm like GS will not be able to leverage the guidance since it wont be multi-view consistent.
- To some degree that the 2D boundary guidance works is surprising since those 2D boundaries often stem from occlusion boundaries where multiview consistency across larger baselines is not given. I would love to see some examples where the 2D boundary is a occlusion boundary and the camera observes it from different angles - does this guidance still work?
- It is unclear why the box-focused sampling cannot re-use the segmentation masks? All that is needed is to assign object probabilities to Gaussian splats which does not need frustra to be unprojected. The GS centers can simply be projected into all images, to assign mask confidence values from Grounded SAM. This would make for a more simple story and system.
- The main results Table 1 incorrectly identifies only NeRF-Det as a baseline for the proposed approach. Other methods like ImGeoNet, CN-RMA, ImVoxelNet also only rely on posed images and are valid baselines for the proposed method. At a minimum each related method needs to have called out the input modalities. As is the table is not useful in providing comparisons to the right kind of related work using also multiview posed images.
- Feature-metric GS/NERF reconstructions such as in LangSplat, EgoLifter, LERF could in principle be prompted for 3D segmentations that of course can be used to extract 3D bounding boxes. Any one of those would be a great additional point of comparison.

**Questions:**

NA

---

### Official Review · Reviewer_fDRh · 2024-11-04

**Soundness:** 3
**Presentation:** 3
**Contribution:** 3
**Rating:** 6
**Confidence:** 3

**Summary:**

The paper proposes to use 3D Gaussian Splatting (3DGS) as the representation to do 3D object detection. To make 3DGS works well for the 3D object detection task, the authors tackle two obstacles: (1) due to the natural of Gaussian, the object boundaries are ambiguous and hard to distinguish, and (2) excessive amount Gaussian blobs in the background. To solve (1), the authors rely on 2D image boundary guidance, and to resolve (2), the authors propose "box-focused" sampling. Experimental results show that the proposed method outperform the current SotA (NeRF-Det) by a reasonable margin on ScanNet and ARKit datasets.

**Strengths:**

+ To my best knowledge, this is the first paper that got 3D object detection working using the 3D Gaussian Splatting representation
+ The performance of the method is good. The proposed method outperforms SotA method such as NeRF-Det by a reasonable margin
+ The presentation of the paper is good. The paper is relatively easy to follow.

**Weaknesses:**

- The proposed method seems to be relying on some heuristics and some hyper-parameters seem to be set to work well for the particular test sets. For example, "Gaussian blobs not belonging to any frustum are assigned a small probability pbg, set to 0.01 in practice." lines (315-316)
- The paper does not discuss anything related to latency. The proposed "boundary guidance" and "box-focused sampling" could be taking more computational time than the baseline NeRF-Det
- The upper-bound performance of the proposed method is bounded by various methods used to provide "boundary guidance" and "box-focused sampling", such as Grounded SAM, Suzuki-Abe algorithm etc. If NeRF-Det is aided by these segmentation method, it could potentially perform better as well

**Questions:**

See weaknesses

---

### Note · Authors · 2024-11-15

**Comment:**

We are grateful to the reviewers, ACs and PCs for your time, comments, and interest in our work. After discussion, we've decided to withdraw the submission this time. We will incorporate valuable suggestions and address misunderstandings in the next version. Thank you once again.

**Withdrawal Confirmation:**

I have read and agree with the venue's withdrawal policy on behalf of myself and my co-authors.